# Could an Anterior Cruciate Ligament Be Tissue-Engineered from Silk?

**DOI:** 10.3390/cells12192350

**Published:** 2023-09-25

**Authors:** Judith Hahn, Clemens Gögele, Gundula Schulze-Tanzil

**Affiliations:** 1Workgroup BioEngineering, Institute of Polymer Materials, Leibniz-Institut für Polymerforschung Dresden e.V. (IPF), Hohe Straße 6, 01069 Dresden, Germany; judith.hahn@ukdd.de; 2Institute of Anatomy and Cell Biology, Paracelsus Medical University, Nuremberg and Salzburg, Prof. Ernst Nathan Str. 1, 90419 Nuremberg, Germany; clemens.goegele@pmu.ac.at

**Keywords:** silk, ACL, fibroin, sericin, ligamentocyte, tissue engineering, *bombyx mori*, scaffold

## Abstract

Silk has a long history as an exclusive textile, but also as a suture thread in medicine; nowadays, diverse cell carriers are manufactured from silk. Its advantages are manifold, including high biocompatibility, biomechanical strength and processability (approved for nearly all manufacturing techniques). Silk’s limitations, such as scarcity and batch to batch variations, are overcome by gene technology, which allows for the upscaled production of recombinant “designed” silk proteins. For processing thin fibroin filaments, the sericin component is generally removed (degumming). In contrast to many synthetic biomaterials, fibroin allows for superior cell adherence and growth. In addition, silk grafts demonstrate superior mechanical performance and long-term stability, making them attractive for anterior cruciate ligament (ACL) tissue engineering. Looking at these promising properties, this review focusses on the responses of cell types to silk variants, as well as their biomechanical properties, which are relevant for ACL tissue engineering. Meanwhile, sericin has also attracted increasing interest and has been proposed as a bioactive biomaterial with antimicrobial properties. But so far, fibroin was exclusively used for experimental ACL tissue engineering approaches, and fibroin from spider silk also seems not to have been applied. To improve the bone integration of ACL grafts, silk scaffolds with osteogenic functionalization, silk-based tunnel fillers and interference screws have been developed. Nevertheless, signaling pathways stimulated by silk components remain barely elucidated, but need to be considered during the development of optimized silk cell carriers for ACL tissue engineering.

## 1. Introduction

The incidence of anterior cruciate ligament (ACL) injuries is increasing [1]. The gold standards for ACL reconstruction are autografts such as Hamstring muscle tendons and allografts [1]. Due to the limited availability of auto- and allografts as well as the donor-side morbidity of autografts, a tissue-engineered alternative would be of high interest for surgeons [2,3]. Despite the development of a lot of different experimental ACL tissue engineering approaches [4,5,6,7,8], a tissue-engineered ACL graft has not entered the clinic before now [5]. The great challenge in ACL reconstruction is the high biomechanical loading in the knee joint, which has to be sustained by the graft. For this reason, biomaterials are needed that fully tolerate high repetitive tension and simultaneously allow for ligament formation, graft integration and stable bony attachment via intimate cell–biomaterial interaction. To fulfil this, biomaterials of natural origin such as silk [9], or of synthetic sources, e.g., poly-L-lactic acid (PLLA) [4,10], were experimentally studied. Silk has a long history as a suture material [11,12] and offers optimal structural properties due to its hierarchical organization (Figure 1); long-term, stability exhibiting a slow degradation in vitro [13] and in vivo [14,15,16]; high biocompatibility after removal of its more immunogenic component, called sericin [5,16,17]; and sufficient biomechanical properties [12]. Due to providing helpful properties, silk is often used as part of composite scaffolds/materials, e.g., with collagen, polyethylene terephthalate (PET), poly(p-dioxanone) (PDS) and poly(L-lactide-ɛ-co-caprolactone) (P(LA-CL)) for tendon and ACL grafts [9,18,19,20], or as a coating by means of a functionalization strategy [18,21,22,23]. It has already, years ago, been implicated in ACL reconstruction [24,25,26]. However, the question arises of how the specific requirements of ACL grafts in regard to cell–biomaterial interaction could be fulfilled by silk scaffolds. The start of a pilot study using a tissue-engineered graft in humans was mentioned by Richmond et al. [27] in 2010. Nevertheless, published results are still not available. Archer and colleagues reported in their survey eight years later that there are only few plans to start a clinical trial [5]. For successful ligamentization of a silk-based ACL graft, the specific response of cell types relevant in ACL reconstruction to silk components has to be discussed in detail.

## 2. Materials and Methods

Mesh terms such as those listed below were used for searches in Medline and Google scholar: silk and ACL (62 hits), silk and cruciate ligament (59 hits) and terms leading to many hits, such as sericin/fibroin and ACL/human ACL, silk combined with tissue engineering, biomechanics, collagen. We searched and cross-checked articles identified with terms such as spider silk and cruciate ligament (no results), 3D printing and ligament (3 hits), ACL and Sharpey (55 hits)/osseointegration (90 hits), clinical trial and ACL (the latter with no results). The search was conducted until 31 July 2023.

## 3. Results

### 3.1. Mulberry and Non-Mulberry Silk Proteins, Sources of Silk

Silk consists of fibrous proteins and is stored in silk-producing arthropods (e.g., silkworms, spiders and flies) as a liquid in specialized glands. Secretion leads to fiber formation via spinning or shearing. Originally, silk was harvested from the cocoons of the raised “domesticated” silkworm (*Bombyx [B.] mori*) and fed with a pure mulberry tree diet (Figure 1). Nowadays, many other natural silk sources have been exploited, such as those derived from wild silkworms, e.g., Antheraea pernyi [30] or diverse spider species (sp.), e.g., the Nephila sp. (clavipes or edulis), Linothele megatheloides, Brachypelma sp. or Araneus ventricosus [31]. Recently, the egg stalks of the green lacewing (Figure 1) have also attracted interest because they contain a biomechanically very stable kind of silk [28]. In addition, natural silks of other insects or even other animals [32,33], as well as diverse silk sources, were investigated [34]. Due to scarcity, especially relevant for spider silks [12] and source-dependent batch to batch variations, recombinant silk proteins have become a valuable alternative [11]. In the case of spider silk, the gene transfer of spider silk genes into *B. mori* [35,36], into bacteria and into other expressing organisms [37,38] was established to achieve an upscaled silk production. Nevertheless, the gene product often showed lower molecular weight and a lack of the typical N- or C-terminal (NT, CT) regions, which can influence biomechanical properties [12,39]. Specific modifications of the terminal regions can change the behavior of cells exposed to silk, such as the differentiation of mesenchymal stem cells (MSCs) [38].

### 3.2. Structure of Silk

The structures of silk derived from silkworms and spiders share many features, such as that the proteins are highly repetitive in their amino acid sequences and soluble at high pH levels. Silk is stored in the animals’ glands as a random coil or α-helix structure, but changes conformation into β-sheets when released by spinning [40]. This conformational change is most often mediated by exposition to a pH gradient during secretion [40]. The parts of silk fibroin (silkworm) or spidroins (spider silk) which remain in a random coil or α-helical conformation are called amorphous parts, whereas the fraction organized in β-sheets is named nanocrystalline [29,41]. The structure of the nanocrystalline fraction of silk fibroin as a linear polypeptide (*B. mori*) has been analyzed in detail: it has a lamellar structure with β-turns every eight amino acids [42]. It contains as dominating amino acids non polar glycine and alanine [43] and comprises different proteins such as light, heavy chains and p25 protein, which are associated with each other and with sericin [40]. Dragline silk consists of so-called spidroins (1 and 2) resembling the *B. mori* fibroin heavy chain due to containing a repetitive sequence with many glycine and alanine amino acids bordered by nonrepetitive N- and C-termini. However, the latter and the repetitive parts differ between spidroins and fibroin [40]. The main categories, silkworm and spider silks, have been directly compared by Yang et al., indicating several differences in the amino acid sequences of their peptides, their secondary structure, and also those in fibroblast interactions with a higher early cell adhesion rate and adhesion forces on the spider silk [31].

The structure and design options of silk scaffolds have been reviewed in detail recently [44]. Similar to collagen type 1, the most abundant protein in the mammalian body and the ACL, silk forms a thread and is most commonly used in tissue engineering approaches. This thread has a diameter of 10–20 µm (*B. mori*) or 3–5 µm (spider dragline fiber) as a multifilament comprising 20 nm diameter nanofibrils of fibroin (*B. mori*) or spidroin (spider) [29]. The latter are in the range of collagen type 1 fibrils (10–100 nm) [45]. Unlike the collagen triple helix, the organized nanocrystalline parts of the nanofibrils of fibroin and spidroins have a β-sheet conformation [29] (Figure 2). The small amino acid glycine, which dominates in the three chains of the collagen triple helix and allows the helix to curl, also plays a major role in silk fibroin and spider spidroin and is part of the RGD recognition sequences of extracellular matrix (ECM) components, known to be important for some crucial cell–ECM interactions [46]. Degradation of fibroin (enzymatically, e.g., by collagenase or proteinase K) was found to depend on secondary structure, indicating that the crystalline part of fibroin (β-sheets) was more stable than the random coil part [41].

In contrast to spider silk, which is composed of 100% spidroin (main protein) with different proportions of amorphous and crystal domains [47], 60–80% of *B. mori* silk consists of fibroin and 15–35% of sericin [48]. Sericin has different rheological properties than fibroin [49] and forms a coating on the fibroin fibers to bind them together [15] (Figure 1). Sericin acts as a viscoelastic shear reducing fluid; thereby, it lubricates the fibroin flow out of the silk-producing glands [49]. Moreover, it influences the bio folding of the fibroin, which means enhanced β-sheet formation. These β-sheets are hydrophobic regions that adsorb proteins and are responsible for attracting inflammatory cells, as reviewed by Ode Boni et al. [50].

The crucial step for processing to obtain a biocompatible silk scaffold is to remove the sericin layer [5]. For further silk processing, it is generally removed [51] either using conventional treatment (soap or alkali treatment), enzymes, CO_2_ supercritical fluid, acids, steam or ultrasonic treatment, known as the so called degumming step [39]. Recently, a self-degumming strategy has been proposed by Wang et al. where a trypsinogen gene is overexpressed in the sericin layer of the cocoons of *B. mori*, leading to digestion of the sericin after the activation of trypsinogen [52]. Nevertheless, despite the interference of processing, sericin also has favorable properties supporting cell adhesion, proliferation and differentiation [53]. In contrast to the more hydrophobic fibroin, it is hydrophilic [15]. Comparing the availability and yield of fibers for spider as well as *B. mori* silk demonstrates major drawbacks of spider silk. Whereas 600 to 1500 m of silk fiber can be obtained from one silkworm cocoon, the fiber yield from a spider gland is only around 137 m. Furthermore, spiders exhibit an aggressive territorial habit and, therefore, are not suitable for series production [32,54].

### 3.3. Biomechanical Properties of Silk

Like collagen, the dominating component of the ACL, silk consists of polypeptides representing a protein. It has a high tensile strength (Table 1) and viscoelastic properties [29]. The strength and stiffness of silk is mainly due to the hydrogen bonds present in the β-sheet crystallites, as well as van der Waals and hydrophobic interactions between the sheets. The ductility is again determined by the semi-amorphous matrix. When subjected to tensile stress, the silk is initially stretched homogeneously until a transition from elastic to plastic deformation occurs, during which the semi-amorphous components dissolve so that the load is then transferred to the β-sheet crystallites until breakage. In principle, silk fibroin can form intramolecular/intermolecular as well as parallel/antiparallel β-sheet crystallites with different sizes and orientations along the fiber axis [55,56,57].

Several studies have been conducted to determine the specific relationship between the amino acid sequence of spider and silkworm silk, its molecular structure and mechanical properties [58,59,60,61]. Also, machine learning was applied to study this interrelation [62,63]. However, its unique properties (Table 1) [29] are in regard to its ultimate tensile strength (UTS) and Young’s modulus, which are superior to cross-linked collagen or even comparable to collagen considering the strain at breakage [15]. Furthermore, silk exhibits an up to ten times higher strength-to-density ratio than that of steel as well as a marked strain hardening behavior [64]. The integration of silk fibroin into knitted or woven polymer scaffolds led to improved biomechanical properties compared with silk-free controls and rat Achilles tendons [65,66].

Nevertheless, one has to bear in mind that properties might change due to the varieties of silk, during silk processing, sericin removal (so-called degumming) and in the presence of cells [67,68].

**Table 1 cells-12-02350-t001:** Dimensions and biomechanical properties of silk and ACL components.

Characteristics	*B. mori*, Silkworm Silk	Nephila Spider Dragline Silk	ACL	References
Fiber diameter	10–20 µm	2–5 µm	Human ACL: 1 cm × 1–4 cm Fascicles/subfascicles: 50–300 μm Collagen type 1 fiber: 0.5–3 µm Fibril: 10–100 nm	[29,69,70,71,72]
Ultimate tensile strength (UTS)	500–740 MPa	1.150–1.750 MPa	600–2300 N Maximum stress: 21–41 MPa, depending on age, sex and health status	[15,71,73,74,75]
Strain/elongation at breakage	4–20%	19–40%	15–30%, depending on age, sex and health status	[15,76]
Young’s modulus	5–17 GPa, depending on with/without sericin, lower with sericin	2–16 GPa	99–129 MPa, depending on age, sex and health status	[15,60,73,77,78,79,80]

### 3.4. Silk Processing for Tissue Engineering

#### 3.4.1. Native Silk

Raw silk fibers of *B. mori* consist of fibroin and sericin, as described before. For further processing, it is necessary to remove the sericin to obtain degummed fibers, which exert a good processability with textile techniques. These silk fibers have been used for centuries as decorative textiles or garments [81], in addition to serving as commercial biomedical suture for decades [82]. Typical textile techniques applied in tissue engineering are braiding and knitting to create an ACL scaffold [25,83,84,85] or in regard to tendon repair weaving [65,86]. Embroidery as a promising textile technique for ACL tissue engineering [87] may be useful for future ACL silk grafts, as has been demonstrated for silk intervertebral disc scaffolds [88].

Natural spider silk can be obtained with special devices (dragline silk) or via harvesting as an egg sac silk, which can be used as naturally or as regenerated silk [89]. This dragline silk was used in nerve and skin tissue engineering [90,91]. Although the mechanical behavior of spider silk is comparable to that of ligaments and tendons, the production of a complete scaffold for ACL tissue engineering seems rather unsuitable due to the small quantities obtained. Irrespective of this, Hennecke et al. demonstrated the potential of a braided spider silk suture for flexor tendon repair with similar tensile behavior to conventional sutures [92,93].

#### 3.4.2. Regenerated Silk Morphologies

Regenerated silk morphologies, such as films, 3D structures, artificial fibers and hydrogels, are an alternative to native fiber structures. In this process, the native silk is brought into solution, which not only changes the structural properties (aligned vs. disordered) [94], but also the resulting mechanical properties (reduction in tensile strength) [81]. Choosing the right solvent is important to modulate the interactions of silk proteins in concentrated aqueous solutions, which often consist of strong acids, ionic liquids and chaotropic salts. The solvents are thought to break the strong intermolecular hydrogen bonds of the β-sheets and denature the protein, but a complete dissolution is still a challenge [32].

As the natural process of releasing silk fibers by *B. mori* and spiders is spinning, electrospinning is a suitable biomimetic technique to produce regenerated silk fiber scaffolds with nanotopology, as reported and reviewed previously [95,96]. Techniques should provide cell carriers with a sufficient porosity for the ingrowth of cells. Electrospinning of silk fibroin with P(LA-CL) was used to obtain nanofibrous scaffolds supporting tendon–bone healing [18]. Porous three-dimensional (3D) structures made via freeze drying or solid form fabrication (3D printing) are ideal for tissue engineering applications to mimic the physiological micro-environment in healthy native tissue. A directional freezing technique was used to prepare regenerated silk scaffolds for tendon regeneration, as demonstrated by Chen at al. [9]. Three-dimensional printing has attracted increasing attention in recent decades to reconstruct injured tissues or to create novel tissues in vitro. In this respect, bioink is essential for a successful process. The silkworm silk can be processed into a bioink, demonstrating a lot of advantages, such as easy structure modification, controlled degradation, cytocompatibility, etc. It allows for numerous cross-linking methods and, therefore, a high amount of different applications are available [97,98]. Silk-based bioprinting inks, biofunctionalized with RGD for tissue engineering, e.g., for wound healing of the skin, were developed [99]. Furthermore, different bioinks for the 3D printing of bone or cartilage scaffolds were investigated combining silk fibroin with gelatin, hyaluronic acid, tricalcium phosphate, hydroxy apatite (HA) or reinforcing silk microparticles with PCL [100,101,102]. However, no studies were found investigating specific 3D-printed silk scaffolds for ACL regeneration, but novel research concepts on 3D-printed organ-on-chip technologies for the ACL are being pursued [7]. Nevertheless, stabilization of the bioink needs cross-linking. Such as that for collagen-containing scaffolds, e.g., with ethyl-3-(3-dimethylaminopropyl)-carbodiimide (EDC) and 1,6-hexamethylene-diisocyanate (HMDI) [8,103], cross-linking is a strategy applied to silk scaffolds to tune their shape and increase their stability. A variety of techniques have been used as reviewed for silk fibroin-containing hydrogels by Farokhi et al. [104]. 

Silk fibroin-coated PET scaffolds intended as ACL grafts were cross-linked with EDC and N-hydroxysuccinimide (NHS), exhibiting high cyto- and biocompatibility [21]. Another study demonstrated that Horseradish peroxidase (HRP)-cross-linking of silk fibroin hydrogel grafts was a suitable chemical cross-linking strategy for tunnel fillers to be applied in the future for ACL reconstruction [105].

Cross-linking is also used for the functionalization and processing of silk fibroin for diverse future applications (Table 2), e.g., with human basement membrane components such as perlecan, using a dityrosine cross-linking technique. This strategy could stimulate endothelial cell adhesion, proliferation and migration [106].

### 3.5. Functionalization Strategies Applied to Silk

Due to the hydrophobic surface and lack of cell adhesion motifs, silk fibroin displaying hydrophobicity in its nanocrystalline zone must be functionalized to improve cell adhesion [15]. These zones promote protein adsorption and increase the likelihood of immune cell recruitment, so it is recommended that a coating with hydrophilic proteins, such as low-bioactive sericin, is applied to mitigate this risk [50,115]. Functionalization should either mimic the natural ECM or offer ECM motifs to promote cell adhesion. Silk harmonizes with collagen and, hence, silk-collagen composites are commonly utilized for ACL tissue engineering [112,113]. Additionally, a study demonstrated that silk fibroin outperforms collagen in terms of fibroblast adhesion [114].

The functionalization of a knitted silk scaffold with nanofibers consisting of a peptide chain (Ac-RADARADARADARADA-CONH2) was performed by Chen et al. to mimic the natural ECM and enhance cell attachment. The expression of tenascin C as a typical fibroblast marker by MSCs cultured on the fibers was higher than that on non-coated controls [84].

Arginylglycylaspartic acid, known as so-called RGD peptide, can enhance integrin binding [46] and, hence, cell–ECM interactions. Therefore, RGD peptides are typical cell adhesion motifs. Arginylglycylaspartic acid peptide functionalization of silk fibroin was conducted to prepare inks for bioprinting [116] derived from spider or *B. mori* silk [99]. The silk of the wild silkworm Antheraea pernyi does already naturally contain RGD sequences demonstrating enhanced corneal epithelial cell growth [30].

So far, no report has been found for ACL scaffolds functionalized with RGD. Recently, genetically designed cytophilic spider silk proteins with cell-adhesive peptide sequences from ECM proteins facilitating cell-specific adhesion were described [117].

Silk combined with collagen (type 1) was used as a scaffold further functionalized with HA added at both ends of the scaffold to optimize osseointegration, mimicking the enthesis zones [26]. In another study, HA was further combined with silver as nanoparticles, providing a functionalization strategy [118] to support osteogenesis and integrate silver as an antimicrobial agent to prevent infections. Despite not being used for silk-ACL scaffolds thus far, bioactive glass could also present an approach for functionalization to improve the bone integration of ACL grafts [119]. It was already combined with silk fibroin for bone tissue engineering stimulating the osteoblastic differentiation of MSCs [120] and in a three-layer scaffold to facilitate wound healing [20,121].

Laponite, a type of clay, can release Ca++ ions and was hence integrated by wet spinning into the silk fibers of a scaffold to facilitate osseointegration [122]. A collagen-silk scaffold releasing stromal cell-derived factor 1 (SDF-1) for ACL repair was designed with a controlled release of SDF-1 for 7 days, improving ligament-derived stem cell recruitment in rabbit knee joints and improving bone tunnel graft healing [112].

Gene immobilization via lentiviral vectors was used as a strategy to create growth factor release (transforming growth factor [TGFβ]3 and bone morphogenetic protein [BMP]2) with triphasic silk scaffolds seeded with MSCs to stimulate an ACL enthesis formation. The enhanced proliferation and lineage differentiation of MSCs as well as the osseointegration of the scaffolds in the rabbit ACL reconstruction model were observed [123,124]. Mesenchymal stem cells were cultivated on ACL scaffolds (made of ultra-high-molecular-weight polyethylene [UHMWPE]) coated with silk fibroin plus vascular endothelial growth factor (VEGF). The VEGF release was maintained for weeks and improved cell proliferation could be seen in vitro, as well as enhanced healing in the rabbit model in vivo [125].

In summary, these strategies reflect diverse opportunities for the cell interaction between fibroin and cells.

Since silk fibroin is generally more cytophilic than many synthetic biomaterials, it was often used to functionalize polymers including PDS, polycarbonate-urethrane (PU), PET and P(LA-CL), respectively [19,20,21,22,43,126].

### 3.6. Cell Response to Silk and Its Components

A recent study by Zhang et al., which focused on skin regeneration, employed multiomics analysis including RNA sequencing of MSCs exposed to silk fibroin and sericin, which indicated that both silk components stimulate generalized but different cellular responses in MSCs. Fibroin and sericin enhance the paracrine functions of MSCs via the regulation of ECM synthesis, angiogenesis and immunomodulation, as well as by activating the integrin/phosphoinositol 3 kinase (PIK3)/proteinkinase B (Akt) and glycolysis signaling cascades (Table 3). The direct comparison of fibroin with sericin revealed higher in vitro and in vivo immunomodulatory effects for fibroin [127].

The silk fibroin coating of PET using a cross-linking strategy enhanced the ligamentization of an ACL graft via elevated cell adhesion, proliferation and ECM deposition [21,22]. Nevertheless, a large animal study revealed no difference in the ACL healing outcome between silk scaffolds implanted with or without cells (in this case, the autologous stromal vascular fraction of adipose tissue), and the difference in the outcome of cell-loaded versus non-loaded scaffolds depended on the time point of analysis [128]. Another study with periodontal ligament stem cells led to better results in scaffolds implanted with cells into rat Achilles tendon defects compared with those without cells [9]. However, a cell-free approach appears to be the most promising means of avoiding the need for two surgical procedures in the knee joint.

### 3.7. Response of Cells to Silk Relevant for ACL Graft Integration

#### 3.7.1. Fibroblasts

A comparison of cell behavior on uncoated PU membranes or those coated with silk fibroin revealed a 2.2-fold increase in early (3 h) and late (30 days) adhesion of human skin fibroblasts by coating [43]. Human fibroblasts revealed no proinflammatory cytokine induction (e.g., tumor necrosis factor [TNF]α, interleukin [IL]-1β, transforming growth factor [TGF]β) on fibroin-coated PU foam scaffolds or membranes, and the collagen deposition on coated and uncoated foams did not differ [43,126]. Moreover, Tsubouchi et al. reported that human skin fibroblasts increased their proliferation on silk fibroin scaffolds [129]. Jiang et al. proved the higher cytocompatibility (increased cell adhesion and adhesion forces, cell covered areas, proliferation and collagen deposition) of PET scaffolds loaded with mouse fibroblasts and coated with silk fibroin compared with uncoated scaffolds [22].

As skin fibroblasts could be useful for tendon tissue engineering [130], they might also be a cell source to support ACL repair.

In addition to fibroblasts, Unger et al. tested the behavior of a broad range of other cell types (endothelial cells, osteoblasts, keratinocytes, glial cells) on silk fibroin nets (*B. mori*) and confirmed a high cytocompatibility of fibroin [131].

The resident cell types within the ACL are specialized fibroblasts known as ligamentocytes, exhibiting comparable behavior to tenocytes, the fibroblasts in tendons.

#### 3.7.2. Ligamentocytes and Ligament-Derived Stem Cells

However, despite representing the dominating cell population in the ACL, only few studies are available culturing differentiated ACL ligamentocytes on silk scaffolds [132,133]. When human ACL ligamentocytes were cultured on knitted silk or silk composite scaffolds, the latter enriched with collagen, ECM synthesis and cell densities were higher in silk collagen composites in comparison with knitted silk scaffolds [133]. Liu et al. directly compared lapine BM-MSCs with lapine ACL ligamentocytes on knitted silk scaffolds. The growth of the BM-MSCs exceeded that of ACL ligamentocytes on the same scaffolds, producing more ligament-related markers. In addition, four weeks after implantation in the rabbit model, more fluorescence-labeled BM-MSCs compared with the ACL cells could be identified, suggesting higher survival rates [132].

The recruitment of lapine ligament-derived stem cells injected into rabbit knee joints via a silk collagen scaffold supplemented with SDF-1 was investigated. Compared with the control group without cells, a superior graft integration became evident [85,112].

#### 3.7.3. Tenocytes and Tendon-Derived Stem Cells

Silk scaffolds were combined with tenocytes for tendon repair approaches. Spidrex, a non-mulberry silk scaffold, was compared with a mulberry silk scaffold, designed to support rotator cuff healing. Tenocytes on spidrex showed adherence, proliferation and tendon-related gene expression, but also features of early immune response [134]. The functionalization of PDS/P(LA-CL) scaffolds using a silk fibroin solution improved the attachment and proliferation of murine tendon-derived stem cells (derived from the patellar tendon); hence, the authors recommended the scaffolds for ACL reconstruction [20].

#### 3.7.4. Synovial Fibroblasts

Another specialized fibroblast cell type of the knee joint localized in the synovial membrane (Figure 2C and Figure 3) and associated with the ACL is synovial fibroblasts (also called synoviocytes type B). However, so far, no published reports could be found concerning their interaction with silk and silk scaffolds. This cell type’s response to silk components has to be investigated in more detail in the future due to its capacity to trigger immune responses and inflammation in the knee joint.

#### 3.7.5. Adipose-Tissue-Derived Stem Cells

The stromal vascular cell fraction from the Hoffa fat pad (sheep model) was suspended in fibrin glue and injected on implanted braided silk scaffolds. This cell population is heterogenous, comprising stem and progenitor cells, pre-adipocytes, endothelial cells, pericytes, T cells and M2 macrophages. Compared with cell-free reconstruction, healing was superior at 6 months post implantation, but not significantly different at the 12 month time point of investigation [128]. A PDS scaffold with silk fibroin possessing a wavy topology with parallel fiber arrangement was seeded with adipose-derived stem cells (ASCs). The fibroin component supported cell attachment, proliferation and phenotype maintenance. A combination of growth factor and mechanostimulation further triggered tenogenic gene expression in PDS/silk composite scaffolds with a wavy topology seeded with human tenocytes [135]. Another study with rat ASCs was conducted on silk fibroin/poly-3-hydroxybutyrate (P3HB) scaffolds with aligned topography and tenogenic differentiation factor-5 (GDF-5) induction. The cells proliferated and differentiated on the cell carrier, but GDF-5 had no further inductive effect compared with the controls without GDF-5 supplementation [136]. Cherng et al. investigated the expression profile of human ASCs exposed to sericin and found elevated synthesis of adhesion-promoting molecules; they suggested an activated transcription of differentiation and migration factors in human ASCs as well as those like PKCβ1, RhoA and RasGFR1, which play a role in wound healing [137].

#### 3.7.6. MSCs

In regard to ACL reconstruction, bone tunnels are created for the graft (Figure 3) and, hence, MSCs play a major role in the osseointegration of the grafts. This is an argument for MSCs being the most often used cell source to seed ACL silk scaffolds. Indeed, lapine bone marrow (BM)-derived MSCs seem to be promising cell candidates for ACL grafts in combination with silk scaffolds [5,14,16]. Lapine BM-derived MSCs were seeded on a knitted silk scaffold functionalized with peptides, and the supplementation with peptides improved proliferation, metabolism and fibroblastic differentiation [84]. Mesenchymal stem cells adhered well on chitosan silk PET scaffolds, and also on those further supplemented with HA with/without silver [118]. Lapine BM-MSCs cultured on knitted silk scaffolds exhibited a high cell proliferation and GAG content compared with lapine ligamentocytes as well as protein and gene expression of ligament-related ECM markers, demonstrating their suitability for ligament tissue engineering [132]. Mechanostimulation was used as a strategy to induce the ligamentogenic differentiation of MSC on a silk fiber mat. The onset of an anabolic cell response was strongly dependent on the time point of stimulation, and mechanostimulation in a bioreactor supported the ligamentogenic differentiation of MSCs [17,138]. Nevertheless, not only mechanostimulation and scaffold topology, such as fiber alignment [139], but also the stiffness of silk hydrogels directed MSC lineage differentiation [140]. This knowledge should be expanded for the specific ligamentogenic induction of stem cells. In particular, the proper adaption of the stiffness of silk-based ACL grafts or even graft zones, e.g., using cross-linking strategies, could favor ligamentogenesis in the midsubstance and chondro-/osteogenesis in the enthesis zones mediated by stem cells in situ without the need for growth factor supplementation to improve graft integration.

#### 3.7.7. Endothelial Cells

These cells are the actors driving angiogenesis. The mature ACL is a poorly vascularized tissue, and the fibrocartilaginous zones of the mature enthesis parts are even void of blood vessels [141,142]; hence, angiogenesis appears, on the first look, subordinary for the ligamentization of ACL scaffolds. However, the graft ingrowth and integration steps into subligamental bones need vessel formation for graft supply. In the beginning of graft integration, a fibrous and immature enthesis becomes evident with Sharpey fibers [16].

#### 3.7.8. Osteoblasts

For osseointegration to restore a functional enthesis osteoblast, activity is required, and silk scaffolds should allow for osteoblast adhesion. Murine pre-osteoblasts of the MC373-E1 cell line were cultured on artificial ligament scaffolds with silk/laponite hybrid fibers and displayed higher cytocompatibility and osteogenic differentiation on the fibers with laponite. Hence, laponite harmonizes with silk and could osteogenically functionalize it [122]. The same cell line was seeded on fibroin/P(LA-CL) electrospun nanofiber scaffolds. The latter showed cytocompatibility with this cell line as well as biocompatibility and higher osseointegration in the rabbit extra-articular reconstruction model compared with the control group (autologous tendons) [18]. Biomimetic composite tubular grafts were made of HRP-cross-linked silk fibroin hydrogels containing ZnSr-doped β-tricalcium phosphate particles and were then tested with osteogenic SaOs-2 osteosarcoma cells, which showed adherence and proliferation on the scaffolds [105].

#### 3.7.9. Chondrocytes

The transition zone of the enthesis between the ligament and bony parts consists of fibrocartilage. Hence, chondrocyte adhesion and interaction with silk is of high interest.

There are few studies which address this fibrocartilaginous interzone of the ACL with silk scaffolds. In the study by Fan et al., gene immobilization utilizing a lentiviral vector associated with TGFβ3 expression was used to successfully trigger the chondrogenic differentiation of MSCs on a stratified silk scaffold [124]. A trilineage co-culture system was applied in vitro based on a silk hydrogel, where BM-MSC successfully exhibited fibrochondrogenic differentiation in the neighborhood with fibroblasts and osteoblasts [143].

In another study, co-cultures on a triphasic silk scaffold were performed with three cell types, including chondrocytes, in addition to BM-MSCs and osteoblasts. The results revealed lineage differentiation of the cell types and successful osseointegration in vivo [144].

These above cited studies and a review of the recent literature with silk and osteochondral repair [145] underline the suitability of silk for chondrogenic differentiation.

#### 3.7.10. Macrophages

Concerning graft remodeling and integration as well as inflammatory tissue response, macrophages play a major role. The study by Cheng et al. revealed a high cytocompatibility of macrophages to sericin exposure. A macrophage phenotype shift to the M2 type was induced, associated with an enhanced expression profile for CXCL9, IL-12A, BMP7 and anti-inflammatory IL-10. Nevertheless, this study targeted proteins important for wound healing [137]. Jo et al. observed that a subfraction of sericin stimulated in RAW264.7 murine macrophage-like cells and human monocyte bone formation [146]. However, for spider silk, a foreign body reaction was observed after implantation into the vertebral canal surrounding a spinal cord lesion [147]. Whether this immune response plays a role in the joint needs to be addressed in future. In regard to *B. mori* fibroin, promising strategies have been proposed to counteract macrophage activation. Accordingly, it was reported that a silk fibroin scaffold modified with MSC-derived ECM could suppress M1-macrophage polarization [148]. Another approach to promote the M2-macrophage shift and reduce foreign body reactions was the coating of a silk scaffold with the anti-inflammatory cytokine IL-4 using click chemistry [149].

### 3.8. Signaling Pathways Stimulated with Silk Components in Cell Types Relevant for ACL Reconstruction

Multiomics analysis was performed with MSCs exposed to silk fibroin and sericin. Both components induced diverse cellular responses and enhanced paracrine activities in MSCs involved in ECM synthesis, angiogenesis and immunomodulation, obviously triggered by integrin/PI3K/Akt and glycolysis signaling pathways. The immunomodulatory capacity in response to fibroin was higher compared with the effect of sericin. However, the focus of this study was on skin regeneration [127].

The study by Ma et al. further underlined the importance of the mentioned signaling cascade activated in response to silk fibroin exposure. In the described experiments, fibroin was combined with HA as a coating to functionalize titanium grafts. It improved rabbit osteoblast adhesion, proliferation and differentiation in vitro, as well as osseointegration, even in a diabetes rabbit model. A specific PI3K/Akt inhibitor antagonized anabolic osteoblast response [150].

The above-mentioned study by Jo et al. showed that sericin activated the BMP2/4 pathway via Toll-like receptor signaling (TRL2, 3 and 4) in murine RAW264.7 macrophage-like cells mediating osteogenic gene expression in co-cultured osteoblasts and osteogenesis in a rat calvaria defect model. In this study, the higher-molecular-weight sericin fraction with a high amount of β-sheet structures was more effective than low-molecular-weight fractions [146].

Murine embryonic multipotent fibroblast-like cells (C3H10T1/2) were treated with silk peptides to assess their influence on adipogenic differentiation. A blocked adipocyte-specific gene expression of peroxisome proliferator-activated receptor γ and its targets, including aP2, Cd36 and CCAAT enhancer binding protein α, was observed. Hence, silk peptides appear to inhibit adipogenesis via the suppression of the Notch pathway, repressing the Notch target genes Hes-1 and Hey-1 [151]. However, to the best of our knowledge, most of the signaling pathways involved in the effects observed during the interaction of silk with cells (Table 3) still remain barely understood and could be strongly influenced by the processing of silk.

**Table 3 cells-12-02350-t003:** Cell responses to silk components.

Components	Cell Type	Effect	Reference
Fibroin	MSC	Integrin PIK3 pathway, immunomodulation	[118,127]
Silk (fibroin)	Ligamentocytes	Lesser effect on ligamentocytes compared with BM-MSCs	[132,133]
Fibroin	Fibroblasts	Support of growth	[131]
Fibroin	SaOs-2 cells (osteosarcoma) MC3T3-E1 (pre-osteoblasts)	Adhesion, proliferation, osteogenesis	[105,122]
Raw silk with SDF-1	Ligament stem/progenitor cells	Cell recruitment	[112]
Fibroin (aligned/random fibers)	Periodontal ligament stem cells (wisdom teeth-derived)	Proliferation	[9]
Fibroin	Endothelial cells	Support of growth	[131]
Fibroin	Stromal vascular fraction	Support of tissue formation in vivo at 6 months	[128]
Sericin	MSC	Regulates glucose metabolism, oxidative stress, angiogenesis, cell adhesion, adaptation to hypoxia and immunomodulation in MSCs, glycolysis and angiogenesis Does not influence gene markers of adipogenic, osteogenic and chondrogenic lineage differentiation, as well as stemness maintenance	[127]
Sericin	ASC	May stimulate the secretion of beneficial adhesion molecules from ASCs and activates the gene transcription associated with differentiation and migration of ASC, regulating regeneration of inflamed tissues	[137]
Sericin/fibroin	Macrophages	Sericin: improved differentiation of macrophages towards the M2 phenotype, ratio of fibroin/sericin determines macrophage phenotype, even the topology of both components elicited different macrophage responses	[137,152,153]

ASC: adipose stem cells; MSC: mesenchymal stem cells; Na: not available, PIK3: Phosphoinositide 3-Kinase, SaOs-2: human primary osteogenic sarcoma cell line, SDF1: stromal-derived factor-1.

### 3.9. Silk in View for ACL Tissue Engineering

As outlined above regarding its overall properties and some similarities to collagen (Figure 2, Table 1), silk has a great advantage for ACL tissue engineering. Furthermore, silk also possesses a hierarchical composition somehow resembling the collagen bundle architecture in the ACL (Figure 1).

The biomechanical behavior of a human ACL under tensile loading typically comprises three phases, the toe-region, linear region and yield region. The crimp pattern of the collagen fibers disappears with fiber stretching at low stresses and, therefore, this feature defines the toe region. With increasing load, the phase in the elastic deformation is referred to as the linear region. The beginning of plastic deformation marks the yield region, which finally continues until failure with an ultimate tensile strength between 600 and 2160 N, stiffness values between 129 and 242 N/mm, Young’s modulus between 99 and 128 MPa and an elongation of 15–30% [78,79,80]. Viscoelasticity is also a crucial functional feature of the ACL [71].

Silk has a long stability in vivo [15], and a high biocompatibility after thorough removal of sericin has been reported [17]. In addition, convincing biomechanical properties [108] (Table 1) explain the current efforts to create a silk graft for the substitution of a ruptured ACL. Taken together, many other advantages and only few disadvantages characterize silk for ACL tissue engineering, as summarized in Table 2.

A review on the in vivo responses of various tissues exposed to silk fibroins (*B. mori*) was performed previously. It describes that fibroin degradation depends on the site of implantation, the method of material manufacturing and processing, extending between a great span of hours to years. Only mild inflammation after a couple of weeks with the recruitment of some macrophages and foreign body giant cells could be observed [109]. Vascularization is possible, but it is rather poor when angiogenic growth factors, e.g., released by cells seeded on it, are lacking [109]. However, the ACL was included in this overview as a graft site and showed similar limited tissue ingrowth, which was improved with MSC pre-seeding [14,109]. Ligament stem/progenitor cell sheets were cultured on knitted silk-collagen scaffolds intended for ACL reconstruction. Scaffolds with and without sheets were ectopically implanted in rabbits, and those with sheets displayed superior results with lesser immune cell accumulation, more fibroblast-like cell growth, ECM deposition and neovessel formation, as well as tissue integration [85].

In another study on a rabbit ACL injury model, a silk-collagen scaffold was tested indicating superior cell immigration, adhesion and graft integration with ligament-related gene expression compared with the control group, and also showed a slow degradation profile in a subcutaneous implantation model [113]. Furthermore, a silk-collagen scaffold with SDF-1 supplementation represents successful integration [112].

A sheep model was used to analyze silk scaffolds for ACL healing, demonstrating an integration into the bone tunnels characterized by the formation of a fibrous connecting interface tissue [6,128]. A porcine model was selected by Fan et al. to investigate a silk mesh (lyophilized silk sponge incorporated in a knitted silk mesh) rolled around a braided silk cord as an ACL graft seeded with MSCs. Sharpey fibers developed which connected the graft with the surrounding bone, and ligament zones became detectable, altogether representing the features of graft integration [16]. Hence, promising results were initially obtained through these large animal models; however, outcomes in a human model are still pending.

### 3.10. ACL Enthesis and Osseointegrating Silk Scaffolds/Devices

Looking at a reconstructed ACL, the highest risk of failure after reconstruction is graft loosening. Therefore, a firm osseointegration is desired, mediated by enthesis zone formation [142]. Hence, silk scaffolds with enthesis zones were generated. Bi et al. used silk collagen scaffolds (cell-free) supplemented at both ends with HA for ACL reconstruction in a rabbit model and found better osseointegration compared with scaffolds without HA [26]. In a rabbit ACL substitution model with different silk fibroin-containing scaffolds (chitosan, PET, HA, silver), bone integration (osseointegration) was analyzed [118]. Bone healing was also demonstrated in a dog ACL reconstruction model using a polyethylene terephthalate (PET)/silk hybrid scaffold [19]. In contrast to PET alone, the silk/PET hybrid supported tissue ingrowth into the scaffold. A chitosan–silk–PET–HA–silver combination showed promising results supporting cell growth [19]. When ACL reconstructions were combined with tibial tuberosity advancements in dogs, silk/MSCs preparations stimulated the bone healing of osteotomy injuries [154]. Silk fibroin collagen scaffolds combined with MSCs stimulated bone integration after ACL reconstruction in a rabbit model [26]. Osseointegration in rabbit knee joints was improved using PET scaffolds with fibroin and HA [23]. A triphasic silk-based scaffold mimicking enthesis zones was tested in another study seeded with three cell types (MSCs, chondrocytes, osteoblasts), where the cells showed characteristic lineage-related phenotypes and osseointegration could be confirmed in a rabbit model [144]. A challenge is to fix ACL substitutes firmly, e.g., by using screws. A novel interference screw for ACL reconstruction with a high content of HA–silk fibroin hybrid particles was tested in a rabbit model and had encouraging biomechanical properties and allowed bone ingrowth [155].

## 4. Conclusions

Due to a multitude of positive properties, silk is an ideal biomaterial for ligament reconstruction. With a long tradition as a textile, several textile techniques, e.g., braiding and knitting, were already used for silk scaffold preparation [84,128]. Although not yet applied for ACL silk scaffolds, embroidery technology could be a future strategy for silk scaffold manufacturing, as has been demonstrated for intervertebral disc scaffolds [88]. The advantage of an embroidered scaffold is the possibility to adapt the biomechanical properties, porosity and pore sizes by selecting a suitable embroidery design, as shown for other materials [87,103]. Furthermore, enthesis zones can be replicated [156]. Embroidery technology and other textile techniques allow one to utilize native, non-degenerated silk fibers. In contrast to synthetic polymer materials, silk variants exhibit superior cell-friendly properties. Hence, they are often used as a coating for these polymers. While many promising results with stem cells cultivated on silk for ACL tissue engineering are available, results with differentiated ACL ligamentocytes or fibroblasts are mostly lacking or even remain rather disappointing. This fact requires a deeper analysis. Interestingly, the signaling pathways triggered by silk fibroin and sericin also remain mostly unexplored. This is probably being caused by the variety of manufacturing and processing techniques applied for silk preparations, modulating the ratios of molecular weight fractions, secondary structures and topologies. These and other unknown parameters influence cell responses to silk. However, candidate pathways triggered by fibroin, such as integrin/PI3K/Akt, and activated by sericin, including the TLR2, 3 and 4 mediated signaling cascades, should be further studied since data for cell types involved in ACL graft healing are lacking. From the biomechanical perspective, silk is an ideal ACL substitute, it is much more cell-friendly than other synthetic grafts already in use and is highly appropriate for ACL tissue engineering. Even sericin, a component which was neglected and removed during degumming for a long time, seems to exert some promising capabilities, particularly on stem cells. It should be explored in more detail in future with regard to its influence on ligamentogenesis.

## Figures and Tables

**Figure 1 cells-12-02350-f001:**
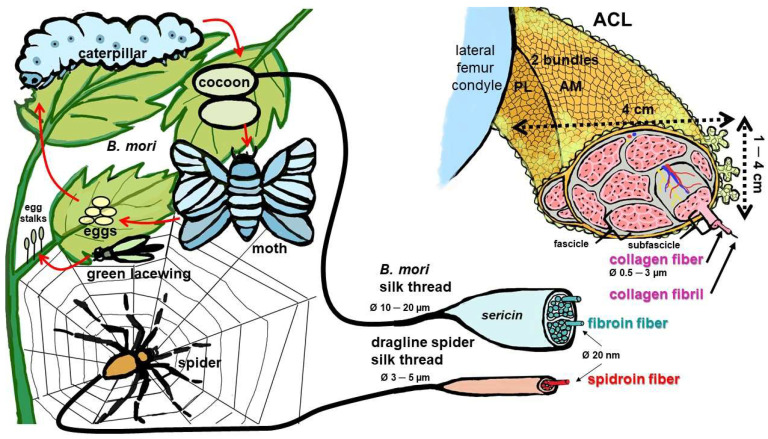
Silk-producing arthropods and hierarchical composition of the silk threads of *Bombyx (B.) mori*, spiders and the anterior cruciate ligament. Life-circle of *B. mori*. Egg-stalks of green lacewings contain silk [28]. Spider net and dragline silk, dimensions as reported by Blamires et al. [29]. ACL: anterior cruciate ligament, AM: anteromedial bundle, PL: posterolateral bundle. Image was created using krita by G. Schulze-Tanzil.

**Figure 2 cells-12-02350-f002:**
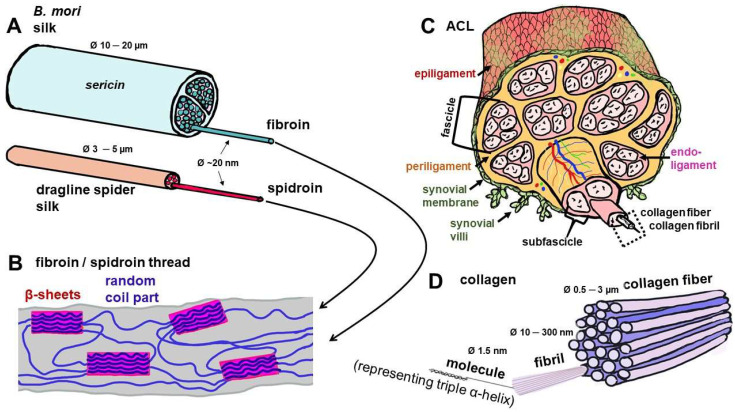
Composition of the silk threads of *Bombyx (B.) mori* and spiders in comparison with collagen as the building unit of the ACL. (**A**) Silk threads of *B. mori* and spider. (**B**) Molecular structure of silk threads with random coil and crystalline parts, the latter containing β-sheets. (**C**) Hierarchical structure of the ACL, mainly consisting of parallel collagen type 1 fiber bundles. (**D**) Detailed structure of collagen fiber. Image was created using krita by G. Brachypelma -Tanzil. ACL: anterior cruciate ligament.

**Figure 3 cells-12-02350-f003:**
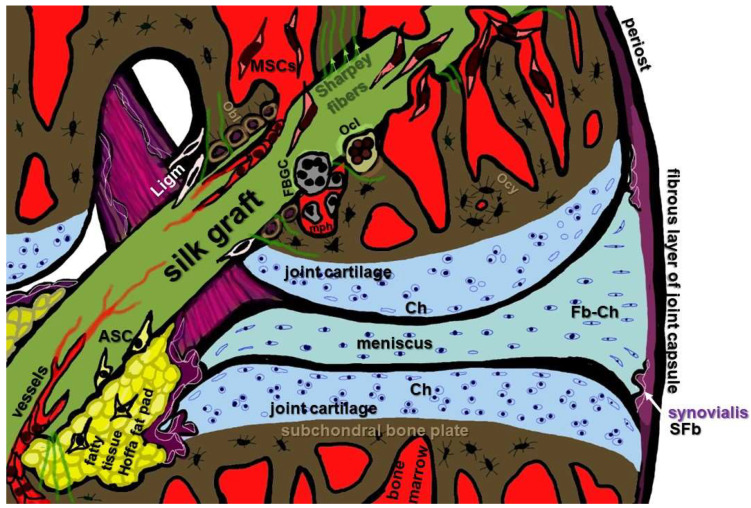
Cell types relevant for in situ graft integration of a reconstructed ACL left knee joint. ASC: adipose-tissue-derived stem cells; Ch: chondrocytes; SFb: synovial fibroblasts; Fb-Ch: fibrochondrocytes; FBGC: foreign body giant cells; Ligm: ligamentocytes from the ACL stump; mph: macrophages; MSC: mesenchymal stem cells; Obl: osteoblasts, Oc: osteoclasts; Ocy: osteocytes; SFb: synovial fibroblasts. Image was created using krita by G. Schulze-Tanzil.

**Table 2 cells-12-02350-t002:** Advantages and limitations of silk as a biomaterial.

Issue	Advantage	Drawback	Reference
Source	Natural	Natural: scarcity, variability, batch dependent	[12]
Costs	Gene technology: recombinant protein expression systems	Natural source: high, e.g., spider silk	[11]
Processing	Versatile: many techniques can be applied, Section 3.4, tunable properties, can be influenced by cross-linking	Processing changes properties	[104,107]
Biomechanics	Stable, suitable for ACL reconstruction	Influenced by processing, etc.	[20,62,108]
Degradation	Long, tunable by processing	Influenced by processing (e.g., sericin removal) and secondary structure	[15,41,109]
Shape stability	High durability, reversible swelling and shrinking		[110]
Preparation, purification	Easy	Sericin removal necessary (immunogenicity) without damage	[5,50]
Conformation	Changeable, fibers	pH-dependent	[40]
Bioadsorbility	Adsorption properties: adsorption of proteins (particularly hydrophobic [beta sheet] fibroin part	Wetting required Adsorption of proteins also leads to inflammatory cell recruitment	[15,50,111]
Functionalization of silk	Multiple strategies possible, Section 3.5	Biomechanics changed	[19,106,112,113]
Cell interaction	Bioactivity		[50,110]
Biocompatibility	If cleaned, high	Higher than synthetic polymers	[17]
Immunogenicity	Fibroin: low	Sericin: high	[111]
Properties	Sericin: hydrophilic	Fibroin: hydrophobic, antimicrobial properties questionable	[15,48]
Sterilization	Heat dry sterilization (180 °C, 30 min): no structural changes in fibroin, no effect on fibroblast adhesion	Autoclaving: structural changes in fibroin	[114]

## Data Availability

Not applicable.

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
