# Peer review of "Could an Anterior Cruciate Ligament Be Tissue-Engineered from Silk?"

_cells, 2023, doi:10.3390/cells12192350_

Round 1
Reviewer 1 Report
In this review, the authors described the potential of silk for ACL application. The basic properties of silk were described well. Also, the advantage and limitation of the silk for ACL application was organized well. Furthermore, the mechanism in ACL regeneration using silk was described. This review is considered beneficial to readers.
Author Response
Nuremberg, 15th September 2023
Dear Ladies and Gentlemen,
Dear Editor,
The authors would like to thank the reviewers for carefully reading the manuscript and their very valuable comments. We modified the manuscript according to the reviewers suggestions with a list of changes shown below. All corrections and addenda performed are indicated in red/underlined in the revised version of the manuscript. Several orthographic and grammatical mistakes were corrected.
We hope you will find this manuscript suitable for publication in the “Cells”. Please do not hesitate to contact me anytime for questions regarding this manuscript.
Sincerely,
Univ.-Prof. Dr. Gundula Schulze-Tanzil
Authors‘ point by point responses to the reviewer‘s comments:
Reviewer 1:
In this review, the authors described the potential of silk for ACL application. The basic properties of silk were described well. Also, the advantage and limitation of the silk for ACL application was organized well. Furthermore, the mechanism in ACL regeneration using silk was described. This review is considered beneficial to readers.
Response: The authors thank the reviewer for evaluating our paper and the helpful comments.

Reviewer 2 Report
The review is well organized and easy to read. A biomechanics approach on tissue engineering is missed. Several aspects of stem cell differentiation may derive from the scaffold biomechanical behavior in the implanted conditions.
The review paper could result more appealing if considering this aspect.
Author Response
Nuremberg, 15th September 2023
Dear Ladies and Gentlemen,
Dear Editor,
The authors would like to thank the reviewers for carefully reading the manuscript and their very valuable comments. We modified the manuscript according to the reviewers suggestions with a list of changes shown below. All corrections and addenda performed are indicated in red/underlined in the revised version of the manuscript. Several orthographic and grammatical mistakes were corrected.
We hope you will find this manuscript suitable for publication in the “Cells”. Please do not hesitate to contact me anytime for questions regarding this manuscript.
Sincerely,
Univ.-Prof. Dr. Gundula Schulze-Tanzil
Authors‘ point by point responses to the reviewer‘s comments:
Reviewer 2:
The review is well organized and easy to read. A biomechanics approach on tissue engineering is missed. Several aspects of stem cell differentiation may derive from the scaffold biomechanical behavior in the implanted conditions.
The review paper could result more appealing if considering this aspect.
Response: The authors thank the reviewer for the positive feedback on our paper as well as the suggestion for improvement. We include now additional literature referring to biomechanical data of tissue engineered scaffolds for ACL substitution supplementing the section 3.3. in section 3.7.6 we added two references showing that mechanostimulation could enhance MSC differentiation on silk scaffolds.
In addition, we integrated this good idea and added the following sequence in section 3.7.6 MSC
“Mechanostimulation was used as a strategy to induce ligamentogenic differentiation of MSC on a silk fiber mat. The onset of an anabolic cell response was strongly dependent on the time point of stimulation and mechanostimulation in a bioreactor supported ligamentogenic differentiation of MSCs [17, 138]. Nevertheless, not only mechanostimulation and scaffold topology such as fiber alignment [139] but also the stiffness of silk hydrogels directed MSC lineage differentiation [140]. This knowledge should be expanded for the specific ligamentogenic induction of stem cells. Especially the proper adaption of the stiffness of silk-based ACL grafts or even graft zones e.g. by cross-linking strategies could favour ligamentogenesis in the midsubstance and chondro-/osteogenesis in the enthesis zones mediated by stem cells in situ without the need of growth factor supplementation to improve graft integration.”

Reviewer 3 Report
This manuscript is a review article that covers the topic of silk and biomedical application, particularly for anterior cruciate ligament repair. Overall, this is a good article and quite comprehensive with good figures and tables but suffers in that in many places it is difficult to read and requires better English grammar. In too many places the article is very choppy, i.e., very short sentences, that makes this manuscript difficult and not pleasurable to read as a review article. This manuscript needs revision.
Specific Comments:
1. The authors should change the title of the manuscript to spell out ACL, anterior cruciate ligament, instead of using the abbreviation. The abbreviation ACL can mean other things.
2. Do not start sentences with abbreviations, there are many throughout the manuscript, for example, page 8, line 286 and page 11, line 397, “MSCs…”, page 12, line 468, “C3H0…” There are others so please correct all.
3. In too many places throughout this manuscript there are very choppy, meaning noticeably short sentences that need to be combined for example, page 2, lines 46-56, page 5, line 177-187, page 6, lines 206-218, page 7, lines 242-250 and many others, please have this manuscript reviewed for this to improve readability. Page 8 consists of very short paragraphs, some one sentence long (page 8, line 266-267) and this needs revision.
4. There are also many awkward phrases and language misuse, for example, too much use of word “like” instead of using the more formal “such as”; page 5, line 172, “…beard in mine…”, unclear, maybe the authors mean “bear in mind”?; page 3, line 114, should be “thread”, not “threat”; page 6, line 230, “Manifold…by [102]”, should cite authors not a number; page 10,line 355, “tenocytes”, the authors should refer to tendinocytes.
5. Page 2, lines 55-56, The start of a pilot study…not available.” There have been follow-up studies by Richmond and colleagues published after 2010.
6. Page 2, lines 65-69. It is not clear why this section is included as a “Materials and Methods section in a review article. In addition, the authors use a small listing of terms, why not use “silk and tissue engineering”, which in a PubMed search lists over 2000 references or “silk and regenerative medicine”, which lists 660 articles and 88 review articles. This is incomplete listing and limited scope.
7. Page 6, lines 225-226, “however, no literature…” is not true as there are over 30 articles on 3D and silk, for example a nice review by Bakieri, A., et. al, Eur. Cell. Mater, 2022,44:21.
8. Page 12, lines 440-447. There is little discussion of how macrophages interact with silk and variants thereof, as well as the important foreign body reaction to silk and byproducts. This important area is neglected and relevant to biocompatibility and degradation.
9. References need to be consistent; some have the journals cited in caps other in lower case, see #1 vs. #2. If more than six authors, please cite first three and then use et. al., if six or less than cite all six authors; references #8 and others are incorrectly cited.
For a review article, this manuscript requires better grammar, as there are too many paragraphs that have very short sentences, just strung together to form a paragraph which makes for a very boring manuscript to read.
Author Response
Nuremberg, 15th September 2023
Dear Ladies and Gentlemen,
Dear Editor,
The authors would like to thank the reviewers for carefully reading the manuscript and their very valuable comments. We modified the manuscript according to the reviewers suggestions with a list of changes shown below. All corrections and addenda performed are indicated in red/underlined in the revised version of the manuscript. Several orthographic and grammatical mistakes were corrected.
We hope you will find this manuscript suitable for publication in the “Cells”. Please do not hesitate to contact me anytime for questions regarding this manuscript.
Sincerely,
Univ.-Prof. Dr. Gundula Schulze-Tanzil
Authors‘ point by point responses to the reviewer‘s comments:
Reviewer 3:
This manuscript is a review article that covers the topic of silk and biomedical application, particularly for anterior cruciate ligament repair. Overall, this is a good article and quite comprehensive with good figures and tables but suffers in that in many places it is difficult to read and requires better English grammar. In too many places the article is very choppy, i.e., very short sentences, that makes this manuscript difficult and not pleasurable to read as a review article. This manuscript needs revision.
Response: We have made extensive orthographical and grammatical revisions optimizing style in the hope of meeting the reviewer's expectations.
Specific Comments:
- The authors should change the title of the manuscript to spell out ACL, anterior cruciate ligament, instead of using the abbreviation. The abbreviation ACL can mean other things.
Response: We changed the title and used the unabbreviated term.
- Do not start sentences with abbreviations, there are many throughout the manuscript, for example, page 8, line 286 and page 11, line 397, “MSCs…”, page 12, line 468, “C3H0…” There are others so please correct all.
Response: We corrected it.
- In too many places throughout this manuscript there are very choppy, meaning noticeably short sentences that need to be combined for example, page 2, lines 46-56, page 5, line 177-187, page 6, lines 206-218, page 7, lines 242-250 and many others, please have this manuscript reviewed for this to improve readability. Page 8 consists of very short paragraphs, some one sentence long (page 8, line 266-267) and this needs revision.
Response: We have revised the manuscript extensively to improve readability.
- There are also many awkward phrases and language misuse, for example, too much use of word “like” instead of using the more formal “such as”; page 5, line 172, “…beard in mine…”, unclear, maybe the authors mean “bear in mind”?; page 3, line 114, should be “thread”, not “threat”; page 6, line 230, “Manifold…by [102]”, should cite authors not a number; page 10,line 355, “tenocytes”, the authors should refer to tendinocytes.
Response: We thank the reviewer for the advice. We have revised the manuscript extensively according to the comments.
As the latest literature search (have a look on the PubMed result list) has revealed, that the term “tendinocytes” is not very common and is no longer used and after consultation with other colleagues in the field of anatomy, the authors believe that the term tenocytes is correct to describe differentiated fibroblasts derived from tendon tissue.
- Page 2, lines 55-56, The start of a pilot study…not available.” There have been follow-up studies by Richmond and colleagues published after 2010.
Response: We did not find a follow-up study by Richmond and colleagues regarding a silk based replacement substitute for ACL reconstruction in human. The published studies are related to ACL reconstruction in general as well as new (allo- or auto-)graft selection techniques and risk factors for ACL graft failure but do not refer to a tissue engineered approach or even a silk scaffold applied for ACL reconstruction. Archer et al., reported 8 years later in 2018 that there are still „few plans to begin human clinical trials“.
We adapted the paragraph in the introduction section:
„a pilot study using a tissue engineered graft in humans has been mentioned by Richmond et al., [27] in 2010. Nevertheless, published results are still not available. Archer et colleagues reported eight years later in their survey that there are only few plans to start a clinical trial [5].“
- Page 2, lines 65-69. It is not clear why this section is included as a “Materials and Methods section in a review article. In addition, the authors use a small listing of terms, why not use “silk and tissue engineering”, which in a PubMed search lists over 2000 references or “silk and regenerative medicine”, which lists 660 articles and 88 review articles. This is incomplete listing and limited scope.
Response: We specified the research terms and added combinations used for the search to prepare an overview especially on silk for tissue engineering of the anterior cruciate ligament. As a result of these search filters, this information was explained in the Material and Methods section for the sake of transparency and comprehensibility for the reader.
- Page 6, lines 225-226, “however, no literature…” is not true as there are over 30 articles on 3D and silk, for example a nice review by Bakieri, A., et. al, Eur. Cell. Mater, 2022,44:21.
Response: In addition to our previous response, a search for "silk + 3d printing + ligament" leads to only three results (2 reviews-one of them is our own, mentioned in the method section now). Regardless, the recommended reference of Bakirci et al. (2022) has now been cited in the manuscript as a future strategy for ACL tissue engineering that could also use silk as a printing material. We integrated this in section 3.4.2:
„However, no studies were found investigating specific 3-D printed silk scaffold for ACL regeneration, but novel research concepts on 3D printed organ-on-chip technologies for the ACL are being pursued [7].“
- Page 12, lines 440-447. There is little discussion of how macrophages interact with silk and variants thereof, as well as the important foreign body reaction to silk and byproducts. This important area is neglected and relevant to biocompatibility and degradation.
Response: The authors thank the reviewer for this very important hind and the authors have added some new information according foreign body reaction on silk scaffolds and strategies to reduce it in section 3.7. (macrophages) with novel literature cited.
“However, for spider silk a foreign body reaction was observed after implantation into vertebral canal surrounding a spinal cord lesion [147]. Whether this immune response plays a role in the joint needs to be addressed in future. In regard to B. mori fibroin, promising strategies have been proposed to counteract macrophage activation. Accordingly, it was reported that a silk fibroin scaffold modified with MSC-derived ECM could suppress M1-macrophage polarization [148]. Another approach to promote M2-macrophage shift and reduce foreign body reaction was the coating of a silk scaffold with the anti-inflammatory cytokine IL-4 by click chemistry [149].”
- References need to be consistent; some have the journals cited in caps other in lower case, see #1 vs. #2. If more than six authors, please cite first three and then use et. al., if six or less than cite all six authors; references #8 and others are incorrectly cited.
Response: We are sorry for the inconvenience and adapted it.
